# Minimal Residual Disease in Breast Cancer: Tumour Microenvironment Interactions, Detection Methods and Therapeutic Approaches

**DOI:** 10.3390/ijms262311346

**Published:** 2025-11-24

**Authors:** Nigel P. Murray, Socrates Aedo

**Affiliations:** 1Consultant Haematologist Hospital de Carabineros de Chile, Santiago 7770199, Chile; 2Faculty of Medicine, University Finis Terrae, Santiago 7501015, Chile; socrates.aedo@gmail.com

**Keywords:** breast cancer, minimal residual disease, tumour microenvironment, relapse

## Abstract

Breast cancer is the most common cancer in women, depending on the sub-type of breast cancer treatment options are different. After completing adjuvant therapy, there are patients who may relapse even many years later. This review examines minimal residual disease, defined as small, microscopic foci of cancer cells that have survived curative treatment, have disseminated to distant tissues, and implanted there. However, the cancer cells do not exist alone but are a small part of the tumour microenvironment, described as an ecosystem. This includes stromal cells, immunosuppressive regulatory T-cells, myeloid derived suppression cells, cancer associated fibroblasts, tumour associated macrophages. The balance of the immunosuppressive tumour microenvironment and the anti-tumour immune response will determine if there is a future relapse. The interactions between the cancer cells and the tumour microenvironment are dynamic and change with time. Most therapeutic options involve therapies directed against tumour cells, only in the last few years has there been attention on the dynamic effects of the tumour microenvironment and the cancer cells on disease progression and the possibility of decreasing the risk of metastatic disease. This article reviews the latest development in preventing metastatic disease by influencing the tumour microenvironment; at best eliminating cancer cells or at least prolonging the latent period of cancer cell dormancy.

## 1. Introduction

Breast cancer is the most common cancer in women around the world [1], with approximately 342,000 women are diagnosed with breast cancer in the USA [2]. This signifies that during a women’s lifetime there is roughly one in eight women who will be diagnosed with breast cancer [2]. Breast cancer has been divided into different sub-types [3]. Those with ductal cancer in situ, those whose tumour cells express the oestrogen receptor (ER) and the progesterone receptor (PgR) with a low proliferation rate as determined by the expression of Ki67 (<14%), known as Luminal A; Luminal B which expresses the ER but not the PgR and has a Ki67 (>14%). ER positivity has been defined as at least 1% of the tumour cells stain positive for the ER using immunohistochemistry, or in other words up to 99% of the tumour cells the expression of ER is not detected. However, the evidence suggests that patients with 1–10% positive for the ER may benefit from endocrine based treatments [4] and a better response as compared to ER negative tumours [5]. Conflicting evidence has been reported that in breast cancer patients with only a 1–10% of cells expressing the ER have more in common, both in the clinical and pathological settings, with ER negative tumours and showed little benefit from hormonal treatment [6,7]. In a retrospective analysis of 411 patients the expression of ER y PgR showed a bimodal distribution. More than 20% of the cohort were negative from ER expression while two thirds had at least 80% of cells expressing the ER. For PgR expression one third were negative while only 38% had an expression of at least 80%. Tumours that did not express ER were of a higher grade in contrast to those tumours with a least 80% ER expression which were of lower grade and earlier T-stage. With increasing ER expression there was a significant decrease in both local relapse and overall survival and a trend towards a decrease in distant relapse [8].

The subgroup of human epithelial growth factor receptor-2 (HER-2) positive breast cancer is defined as at least 10% of the cancer cells express HER-2. This is based on the use of immunohistochemistry; cancer cells which stain with an intensity of 3+, independent of the expression of ER and PgR are classified as HER-2 positive. Those tumours which only show a 1+ staining intensity are classified as HER-2 low breast cancer and tumours which do not express HER-2 as HER negative.

Tumours expressing HER-2 with an intensity of +2 must be reassessed using FISH. So, in other words even in HER-2 positive breast cancer patients there are potentially up to 90% of the tumour cells which are not stained with a 3+ intensity stained. HER-2 positive breast cancer patients form between 10 and 20% of all breast cancers [9].

Triple negative breast cancer (TNBC) is self-explaining, forming between 10 and 20% of breast cancer cases, in which none of the three receptors are expressed. Even so, each subtype of breast cancer shows intra-tumoral heterogeneity with differing subpopulations of tumour cells. Some are resistant to treatment, others may be able to disseminate to the circulation, survive the shear forces and implant in distant tissues. It has been reported that there are two cooperative mechanisms. Firstly, clonal heterogeneity which is defined by the variation in the different phenotypes and thus biological properties and are influenced by spatial and temporal factors [10,11]. The clonal evolution of dominant or resistant breast cancer cells is one potential mechanism for tumour dissemination and/or resistance to treatment [12,13]. However, breast cancer is complex, with diverse cell types and different biological properties even within the same tumour or tumour subtype. This implies that not all subtypes of breast cancer are the same and may require different treatments. More recently, RNA sequencing (scRNA) methods of single cell analysis have confirmed this complex cellular heterogeneity [14,15]. In studies of metastatic breast cancer which analysed the types of cells in the Tumour Microenvironment (TME) have reported that both the immune and stromal environments are highly heterogeneous [16,17]. This dynamic and complex interplay between cancer cells and the TME have a critical role in the dissemination, implantation and development of micro-metastasis [18]. Using scRNA sequencing nine sub-clusters of breast cancer stem cells were reported to have the potential to form micro-metastasis in the draining lymph nodes [18].

## 2. The Tumour Microenvironment

Cancer cells do not exist alone but form an integral part of the tumour micro-environment. From the initial stages of breast cancer, cancer cells interact with the TME to cause an immunosuppressive environment. This TME is composed of host cells, fibroblasts, immune cells, blood vessels, endothelial cells, and the extracellular matrix. Paget, in 1891, proposed the hypothesis of the idea of the soil and seed. The seeds (cancer cells) which fell on stony ground would not germinate or proliferate, while those that fell on fertile ground would germinate, proliferate, and eventually cause metastatic disease [19,20]. The interactions between the different TME components are dynamic and change with time. This multi-factorial process has been described as an ecosystem, classifying it as a “multi-dimensional, spatiotemporal unity of ecology and evolution” [21,22]. In this ecosystem there is an intraspecific relationship with communication between cells and an interspecific relationship between the cancer cells and host factors. It has further been described that this ecosystem represents the total of the primary, regional, distal, and systemic “onco-spheres”. Each with its own local microenvironment, niches and immune, nervous and endocrine systems [23]. These ecosystems or onco-spheres may change with time or because of treatment for the primary tumour. At what stage this process in the metastatic cascade occurs is not known, core biopsies are only performed when there are imaging studies which raise the suspicion of a possible breast cancer. Without a positive image there are no tissue samples to analyse this process. The detection of mammaglobin expressing cells have been detected in the circulation immediately prior to core biopsy, even in intra ductal carcinoma in situ [24]. Sanger et al. [25] reported that even in ductal carcinoma in situ disseminated tumour cells could be detected in the bone marrow. This suggests that this ecosystem is functioning even before a suspicious lesion is detected on imaging studies.

The immunological response to tumour cells comprises two elements that are responsible for eliminating cancer or abnormal cells. The innate immunological response is mediated by natural killer cells (NK-cells) which have cytotoxic properties to eliminate cancer cells. The second is the acquired immunological response, this involves cytotoxic T-lymphocytes (CTL) aided by dendritic cells which present neo-antigens to the CTL and thus enhance the immune response. Inversely, there is the inhibitory side of the immune system, which inhibits these effector cells. The main cells involved are CD4, FOXP3 positive regulatory T-lymphocytes (Tregs), myeloid derived suppressor cells (MDSC), tumour associated macrophages [TAMs] and the cancer associated fibroblasts [CAFs] all having a crucial role to play. These changes in the immune environment are related to cancer progression and the development of resistance to treatment [26]. In theory, more aggressive breast cancer subtypes should produce a more immunosuppressive TME. Agohozo et al. [27] reported that the immune cell composition of the TME was dependent of the subtype of breast cancer, with a higher proportion of CTLs being present in less aggressive breast cancer subtypes. This onco-sphere is complex, dynamic, changes with time as a result of treatment and disease progression. The TME shows multiple interactions between immunosuppression and the immune defence systems. To gain a better understanding of these components it is necessary to describe the role of each individual TME component.

This not also affects the TME of the primary tumour, but the immunosuppressive nature of the pre-metastatic niche, that is one without tumour cells, creating a fertile soil for the future implantation of circulating tumour cells to form the metastatic niche.

## 3. Cellular Components of the Tumour Microenvironment

### 3.1. Cancer Associated Fibroblasts

Cancer cells interact directly with the stromal fibroblasts or via exosomes to change them into cancer associated fibroblasts (CAFs) [28]. This communication between tumour cells and normal fibroblasts converts the normal fibroblasts into CAFs. The interplay between the tumour cells and CAFs results from components of the ECM and intergrins [29]. There are also paracrine interactions through exosomes via growth factors such as transforming growth factor beta (TGF-β) and platelet derived growth factor (PDGF) [30]. This reprogrammes normal fibroblasts into CAFs by the activation of TGF-β1 signalling causing phosphorylation of SMAD proteins. These proteins translate to the fibroblast nucleus and activate the expression of CAF-related gene. The release by tumour cells of PDGF causes the proliferation of CAFs and also acts on normal fibroblasts and activates tumour specific PDGFR signalling [31]. This interaction between the tumour cells and CAFs plays a significant role in the high metastatic potential of TNBC [32,33]. In a mouse model, the release of exosomes that contain miR-370-3p favours the activation of normal fibroblasts to CAFs [34]. While direct physical interactions between tumour cells and CAFs in a matrigel 3D environment released soluble accelerating factors including matrix metalloproteinases (MMP) and MMP-2 [35].

CAFs also produce other important roles in the metastatic cascade: the phenotypic characteristics of the CAFs and activation is mediated by various stimuli including the resident tumour cells and chemotherapy. Fibronectin forms an important part of the ECM; its function is to regulate cell signalling causing conformation changes in both cell-to-cell adhesion and the binding of growth factors. Cancer cells are unable to produce this; they hijack the CAFs to upregulate fibronectin, but this also enables CAFs to secrete cytokines and chemokines enabling them to remodel the TME [36], affecting cancer cell proliferation, invasion metastasis, and angiogenesis [37,38,39]. CAFs are heterogeneous in nature producing differing activation patterns; they attract pro-tumorigenic myeloid cells such as MDSC macrophages and dendritic cells thus facilitating invasion and metastasis via the epithelial–mesenchymal transition, cancer cell proliferation, and resistance to cell death, which facilitates cancer cell dissemination and promoting the metastatic cascade [39]. Finally, CAFs cause a stiffening of the extracellular membrane impeding the infiltration of anti-tumour cytotoxic T-cells and Natural Killer (NK) cells into the primary tumour. The effects of CAFs on the TME has recently been extensively reviewed by Kennel et al. [38]. Both breast cancer cells and CAFs increase the expression and activation of MMP-2 which has a pivotal role in breast cancer progression and relapse. MMP-2 is a type IV collagenase; its expression in breast cancer cells is associated with a worse prognosis [40]. Not all breast cancer cells express MMP-2; those which do are able to pass through the basement membrane and the extracellular matrix and enter the circulation. However, cells that are MMP-2-negative may disseminate in a form of Indian file through the tract left by MMP-2-positive cells [41]. The multiple interactions between the tumour and stromal cells causes a re-programming of the phenotypic characteristics of both components [41,42]. MMP-2, in degrading the extracellular matrix, triggers the proteolysis of cytokines and their respective receptors such as tumoral necrosis factor receptor R, interleukin 6R and 2R [43]. In addition, MMP-2 causes TH2 polarisation of macrophages from the TH1 subtype, therefore restricting the anti-tumour immune response. It is also able to cleave the interleukin-2-R-alpha receptor which, in doing so, suppresses the proliferation of cytotoxic T-cells because of increased apoptosis [44]. MMP-2 in addition to causing local immunosuppression also causes immunosuppression at distant sites by the release of exosomes into the circulation. Exosomes are released by a process of exocytosis from breast cancer cells in the primary tumour. They are membrane-bound vesicles with a diameter of approximately 50–100 nm. They, as previously mentioned, transport different bioactive components that form part of the metastatic cascade, which in part causes the transformation of normal fibroblasts into CAFs by exosomes released from the cancer cells. The exosomes are released into the circulation being secreted by both cancer cells and CAFs. They carry within their membrane, proteins, lipids, DNA, RNA, and enzymes such as MMP-2 [45]. Their function is not only to cause immunosuppression within the primary tumour but are also importantly in the formation of an immunosuppressive pre-metastatic niche for the future implantation of CTCs. These exosomes have been defined as the key drivers of immunomodulation in patients with breast cancer. The extracellular exosomes also alter CTL activity and as such affect the escape of cancer cells from immunomodulation [46]. There is a differential expression of integrins on the exosomes membrane which may play in part in the organotropic distribution of metastasis in different subtypes of breast cancer [47]. The circulating exosomes fuse directly with the plasma membranes of their target cells, releasing their contents into the normal cells and thus occupy the distant TME [48]. This is the pre-metastatic niche devoid of cancer cells but creates a fertile soil for the future implantation of circulating tumour cells. Exosomes that contain miR-105 can disrupt the endothelial cell junctions thus facilitating their invasion of distant tissues [49]. Depending on the breast cancer subtype, the exosome membrane expresses different ligands; intergrin α6β1 and α6β4 are associated with lung metastasis, while αvβ4 with liver metastasis [47]. They also promote bone metastasis in breast cancer by the transfer of miR-21 to osteoclasts [50].

The exosomal contents cause immune dysfunction; firstly, they cause the differentiation of bone marrow monocytes into Myeloid Derived Suppressor Cells (MDSCs), causing local immunosuppression [51]. Tumour cells are able produce chemokine CCL2 [52], which attracts MDSCs, Tregs, tumour associated macrophages and cancer associated fibroblasts into the primary tumour. It also increases the resistance of primary tumour cells against apoptosis and increases their proliferation and migration. It also interferes with the function of cytotoxic T-cells, NK-cells and dendritic cells and is associated with an increased proliferation of Tregs. This worsens the prognosis of the patient by enhancing the immunosuppressive of the primary TME [52,53,54]. These immunosuppressive cells also increase the conversion of immature B-cells into regulating B-cells (Bregs) causing further immunosuppression [55]. This not only affects the TME of the primary tumour but the immunosuppressive nature of the pre-metastatic niche, which is the one without tumour cells, which as previously mentioned creates a fertile soil for the future implantation of circulating tumour cells.

### 3.2. Tumour Associated Macrophages (TAMs)

TAMs may form up to 50% of the total of immune cells infiltrating the primary tumour, causing, throughout different mechanisms, progression of the tumour [56,57]. There are basically two subtypes of macrophages. The activated M1 subtype, which exerts a killing function of cancer or infected cells. The M2 sub-type, however, supports the progression of the tumour by promoting cancer cell invasion, metastasis, angiogenesis, the formation of cancer stem cells and increasing the immunosuppressive nature of the TME. They play an important role in the formation of the pre-metastatic niche and secrete various metalloproteinases, including MMP-2 [58]. They also upregulate the expression of programmed cell death protein 1 (PD-1), which decreases the effectiveness of cytotoxic T-cells of eliminating tumour cells [59,60]. TAMs and MDSC cells cause immunosuppression in the TME in a cell-contact-dependent manner producing a decrease or ineffective T-cell response [61]. This has been reviewed recently in greater depth by Huang et al. [62]. Chemokines play an important role in the crosstalk between tumour and tumour-associated macrophages in the development of the immunosuppressive environment of the primary tumour. Creating therapies that antagonise this process may be one way of overcoming this immunosuppressive effect [63].

### 3.3. Cytotoxic CD8^+^ T-Cells

These cells are an essential part of the host immunological response to eliminate cancer cells. Antigen presenting cells interact with the cytotoxic T-cells to increase their activation and proliferation. This allows the T-cell to bind to the cancer cell in order to eradicate it; however, for this mechanism, the T-cell has to bind to the cancer cell. Thus, the stiffening of the extracellular membrane by CAFs may impede the entrance of CTLs into the primary tumour and thus limit their efficacy [38]. As previously mentioned, tumour cells can inhibit the function of cytotoxic T-cells. The use of antibody conjugates linked to cytotoxic T-cells may improve their efficacy by binding directly to the tumour and will be reviewed later in the article [64].

### 3.4. Natural Killer Cells

These are part of the innate immune system eliminating infected and tumour cells by phagocytosis. As with cytotoxic T-cells the primary tumour can suppress their function, as previously mentioned.

The understanding of how tumour cells escape from the immune system by causing immunosuppression may help to develop new therapeutic agents. Each breast cancer subtype has a different TME and thus requires different treatment approaches. The more aggressive the subtype of breast cancer the more the TME exerts an immunosuppressive effect. In an article by Moura et al. [65], they addressed the question of why Luminal B breast cancer causes a more increased immunosuppressive TME than Luminal A type breast cancer. The report concluded that Luminal B type breast cancer had a higher percentage of Tregs, with lower levels of NK-cells and cytotoxic T-cells, which may affect the patient’s prognosis and a possible development of targeting specific components of the TME.

This understanding is critically important if treatments of minimal residual disease are to be successful and is summarised in Figure 1.

## 4. Dissemination of Tumour Cells to Distant Tissues

Tumour cells that escape from the primary tumour into the circulation must survive the shear forces in the blood and evade immune destruction. They express MMP-2 on the cell membrane permitting them to enter the pre-metastatic niche, converting it into the metastatic niche. Once there, they interact with the new TME causing changes in their biological properties. Most of these micro-metastases become MMP-2 negative and this has recently been reviewed [66].

Circulating tumour cells target the pre-metastatic niches, which according to Paget provides a fertile soil for their future proliferation. The immunosuppression environment facilitates the survival of the tumour cells, converting the pre-metastatic niche into the metastatic niche; these cancer cells can be detected in the bone marrow.

How this down regulation of MMP-2 in the metastatic niche is achieved is not understood.

Theoretically, Tissue Inhibitor of Metalloproteinase 1 (TIMP-1) may inhibit the ability of MMP-1 from converting the zymogen MMP-2 into its active form by affecting the cysteine switch [67] or MMP-2, which is directly inhibited by TIMP-2. The cysteine switch is the removal of a cysteine residue that protects the active site of MMP-2 being activated and thus causes down regulation of this enzyme. However, the loss of activated MMP-2 expression the tumour cells interacts with the components of the new onco-sphere causes the tumour cells to go into a period of dormancy or latency.

During this time there is no net proliferation of the tumour cells, growth and the formation of macro-metastasis. The length of the period of dormancy varies between different types and subtypes of cancers.

Initially, the circulating tumour cells, after leaving the circulation, enter the perivascular niche composed of perivascular cells and sinusoidal endothelial cells [68]. When the breast cancer cells enter the perivascular niche, they transform into breast cancer stem cells. This change to a stem like phenotype is mediated by the Wnt-β-catenin pathway [69]. Here, they interact with exosomes released by mesenchymal stem cells that induce cycling quiescence, DNA repair, and the formation of different subtypes of breast cancer cells [69]. Later, exosomes cause dedifferentiation of the breast cancer cells into a more homogenous stem cell population [69]. These cancer cells remain viable adapting to their new TME and by various mutations and epigenetic modifications are resistant to chemotherapeutic agents and immune surveillance [69]. From there the cancer cells migrate to the endosteal niche composed of osteoclasts and osteoblasts [69]. The breast cancer cells in the bone marrow reside in specific bone marrow niches that can regulate their dissemination to and from the bone marrow [69]. These cancers act as a reservoir for the future development of metastatic disease [69]. The relatively hypoxic state of the bone marrow promotes the change to a stem cell like phenotype and dormancy [70]. Cancer stem cells are thought to be more “dormant” than non-stem cell like cancer cells and are more resistant to the actions of chemotherapy [71].

In the metastatic niche the breast cancer cells may express a different pattern of the receptors, ER, PgR, and HER-2. It has been reported that there is a significant discordance between the expression of HER-2 in the primary tumour as compared to its expression in the metastatic niche [72]. This discordance between the primary tumour and metastatic niche for the expression of the receptors ER, PgR, and HER-2 may be as high as 31.5% of breast cancer patients [72]. This discordance of the expression of HER-2 reached 49% and is associated with a worse prognosis [73]. Konig et al. [74] reported that in a study of 29 patients with Luminal A breast cancer, cells expressing HER-2 were detected in 68% of patients while 76% of cancer cells expressed Ki-67 in bone marrow aspirates. In a meta-analysis of the discordance between ER, PgR, and HER-2 status of the primary tumour, the pooled proportions of tumours changing from positive expression of these biomarkers and the reverse were 24% and 14%, respectively, for the expression of ER, 46% and 15% for PgR, and 13% and 5% for HER-2 [74]

Ditsch et al. in a small study of 17 patients reported that while ER was detected in 65% of primary tumours, only 12% were ER positive in the metastatic niche. They also reported that the expression of ER in the tumour cells found in the metastatic niche was heterogeneous revealing both ER positive and negative cells [75]. The authors concluded that there could be selective dissemination of ER negative tumour cells in the bone marrow or a negative impact of the TME on ER expression. It has been suggested that in the metastatic niche, plasticity of receptor expression may change receptor status and thus have clinical implications on therapeutic options [75,76].

## 5. Subclassification of MRD

It has been reported that there are different types of MRD in prostate and colon cancer [77,78,79]. The authors divided patients into those who were negative for both micro-metastasis in the bone marrow, those only positive for bone marrow micro-metastasis, and those who were positive for CTCs independent of whether bone marrow micro-metastasis was present or not. In stage III colon cancer patients treated with FOLFOX adjuvant chemotherapy, these three subgroups had differing times to relapse and the frequency of patients who relapsed [79]. This action of chemotherapy is to eliminate proliferating tumour cells; however, these survival curves suggest that in micro-metastatic disease chemotherapy is less effective. In this study, those patients with only bone marrow micro-metastasis had a similar survival curve as those patients negative for MRD. However, two years after completing chemotherapy, there was an increased relapse rate suggesting that during the latency period cancer cells are resistant to the effects of chemotherapy due to their non-proliferative state.

In patients with breast cancer, there are few comparative studies of the detection of bone marrow micro.metastasis and CTCs. In a study by Molloy et al. [80] of 733 early-stage breast cancer patients, they reported that the value of the detection of bone marrow micro-metastasis and mCTCs may differ between patients depending on their biological properties. The study had a median observation period of 7.1 years for disease-free progression and 8.3 years for overall survival. The presence of both micro-metastasis and CTCs had a worse prognosis than those patients with only one positive parameter.

Patients positive for CTCs were more likely to have bone marrow micro-metastasis, which seems logical as bone marrow micr-. Based on the study finding, the authors suggested that bone marrow micro-metastasis remain dormant for many years in contrast to CTCs, which represented dissemination into the circulation in more rapidly proliferating micro-metastasis. The authors also suggested that micr-ometastasis has a higher frequency of dormancy and thus is responsible for later relapse. In an earlier study of 60 patients, CTCs were found to be more frequently detected in progressive disease rather than “stable” disease or remission, with a positive correlation of CTCs being detected in those patients with bone marrow micro-metastasis [81,82].

## 6. The Detection of Minimal Residual Disease in Breast Cancer, What It Means and How Can It Be Used to Direct Treatment Options in Non-Metastatic Breast Cancer Patients

In terms of MRD, there are three methods used for their detection: the presence of tumour cells in bone marrow samples, the detection of CTCs in the blood stream, and finally, the detection of circulating tumour DNA (ctDNA). It is important to understand what these three markers MRD signify and how they may be used to direct therapy before the appearance of macro-metastasis.

### 6.1. Bone Marrow Micrometatasis

The analysis of the bone marrow can detect the presence or absence of micro-metastasis; however, it must be understood that a negative result may be due to sampling error and does not exclude the presence of visceral micro-metastasis. It has been suggested that not all cells detected in bone marrow aspirates are micro-metastasis and may be CTCs passing through the bone marrow compartment [83]. Although, evaluation of the bone marrow is more invasive than a simple blood test, the frequency of adverse events was low—16/13,147 (0.08%) of those which were reported, with 11/16 being haemorrhages [84]. This raises the question that while cancer treatment programmes are based on the clinical and pathological findings of the primary tumour, those patients with MRD detected in the bone marrow found in the bone marrow after curative therapy may require a different treatment as the surviving tumour cells may have differing biological characteristics when compared to the primary tumour. Micro-metastasis can be classified using immunocytochemistry as being positive or negative for the expression of HER-2, ER, PD-1, PD-L1, Trop 2, CTLA-4, and PI3K, thus aiding the decision for treatment options. Using serial samples, the evolution of the micro-metastatic disease can be monitored in terms of changes in their biological properties and response to treatment. An increase in the expression of Ki-67, a marker indicating cellular proliferation, may indicate the end of the latency period, although this is just a hypothesis. As previously mentioned, a HER-2 positive micro-metastasis may occur in HER-2 negative primary breast cancer specimens, and this may partially explain why trastuzumab may produce positive benefits in patients classified as HER-2 negative [85]. If there is no or little response, dual HER-2 blockade can be considered such as lapatanin, which inhibits both HER-1 and HER-2 tyrosine kinases. The expression on HER-2 on the cancer cells is dynamic and may change with time, converting HER-2 positive cells into HER-2 negative cells or vice versa [86]. Trastuzumab also affects the immunological equilibrium, increasing antibody dependent cell cytotoxicity via an increase in the activity of NK cells [86]. Those primary tumours positive for the ER undergo treatment with tamoxifen or an aromatase inhibitor such as anastrozol or letrozol; in a similar fashion to HER-2, monitoring the results may in case of failure indicate a change in treatment. It is possible to analyse single cell characteristics of bone marrow cancer cells using immunocytochemistry, the polymerase in chain reaction (PCR) and next generation sequencing (NGS) to further classify bone marrow micrometastasis obtained from aspirates. There are no studies of TLA-4 and PD-1, PD-L1, Trophoblast agent 2 (Trop 2), CTLA-4, and PI3K expression and treatment outcomes in bone marrow micrometastasis. Theoretically, serial measurements of these markers could monitor the evolution of bone marrow micro-metastasis and alter treatment options. A pooled analysis of 4703 patients with stage I-III breast cancer reported that the presence of bone marrow micrometastasis was associated with a poorer prognosis [87].

### 6.2. Circulating Tumour Cells

CTCs represent another tool for detecting MRD, their presence indicating that the latent period is over and tumour cells are disseminating to other tissues. The only FDA approved detection method is the Cellsearch^®^ system (Veridex Corporation, Warren, NY, USA) in patients with metastatic breast cancer. Due to the low frequency of CTCs in the circulation there must be an enrichment process, for which there are several methods each with its own pitfalls. Detection methods can be divided into those which use the biological properties of the CTCs and the other on the physical properties. Density gradient centrifugation is one such method of detection using the biological properties of the CTCs, as they are less dense and thus are separated from other blood components. The CTCs can be further analysed by immunocytochemical methods. CTCs can also be isolated based on their size of epithelial tumour cells this method also allows the immunocytological and molecular characterisation of the CTCs. Methods include involving the physical properties of CTCs using magnetic beads labelled with antibodies which target specific antigens.

This may be achieved using negative selection whereby CD45 positive leukocytes are removed from the blood sample or positive selection whereby CTCs are detected using antibodies against epithelial markers and the different pitfalls of each method has recently been reviewed [88]. Neither the detection of micro-metastasis nor CTCs have been incorporated into clinical guidelines. However, CTCs can be sub-classified equally as can micro-metastasis, for the expression of HER-2, ER, the expression of CD47 and PD-L1.

These biomarkers expressed in CTCs are associated with disease progression, whilst detecting variants of PIK3CA can indicate resistance to HER-2 directed therapies [89]. Similarly, detection of the oestrogen receptor variant 1 has a significant role in the development of resistance to hormonal therapy and can detected by liquid biopsy [90]. The first description of CTCs in metastatic breast was by Ashworth in 1869 [91], but it was not until 2005 that the detection of CTCs was shown to be a prognostic factor in patients with newly diagnosed metastatic breast cancer [92]. A pooled analysis of CTC status in patients with metastatic breast cancer, provided similar results [93]. The PREDICT pooled analysis [94], reported a strong association between repeated CTC determination and overall survival independent of the breast cancer subtype and previous treatment. However, the results of using CTCs to decide treatment strategies have been conflicting. A retrospective analysis of metastatic breast patients showed that a cut-off point of at least five CTCs/blood sample at baseline could enable a method of risk stratification in patients with metastatic breast cancer. It was reported that this cut-off point could be used to classify patients into stage IV indolent cancer, that is with a number of CTCs < five cells/sample and stage IV aggressive cancer. Those patients with an aggressive subtype had a worse prognosis independent of breast cancer subtype or prior treatment [95]. However, the reported results of clinical trials have been contradictory. The STIC CTC trial reported that in a sub-analysis those patients with a high CTC count benefited from chemotherapy compared to those treated with hormonal therapy [96,97]. In contrast the SWOG SO500 trial failed to show an improvement in overall survival in continuing the same chemotherapy regime or switching to an alternative. In patients who had received at least two prior chemotherapy regimens failed to show an improved outcome in the CirCe01 trial [98]. In the DETECT III trial the phenotypic expression rather than the numbers of CTCs detected was compared. Patients with HER-2 negative metastatic breast cancer were compared with patients CTCs HER-2 positive were randomised to either standard treatment or randomised to receive lapatinib, a tyrosine kinase inhibitor. It was reported that the presence of HER-2 positive CTCs the use of lapatinib improved the overall survival compared with the expression of HER-2 in the metastatic tumour [99]. The conversion of the molecular subtypes of CTCs is approximately 82–90% between the primary tumour and CTCs, with the most frequent pattern changes being the conversion to unfavourable breast cancer variants. This may be partially explained by CTC heterogeneity and selection of specific CTC phenotypes and an indication of CTC plasticity. As most CTCs are cleared from the circulation within one to two hours, repeated blood sampling may be needed in patients testing negative for CTCs [99,100]. Trapp et al. [101] determined the presence of CTCs before and after adjuvant chemotherapy; this was part of the phase III Success Trial. Two different chemotherapy regimes were used and after completion zolondrante was added for two years versus five years of bisphosphonate therapy. The presence of CTCs was associated with the patient’s prognosis; two years after chemotherapy, of the 1087 women included in the study 18% had detectable CTCs, which was a prognostic factor for a shorter disease-free survival and overall survival. It was further reported that in this trial of hormone receptor positive patients, approximately 50% suffered a relapse with a latant period of at least five years. Sporano et al. [102] reported that CTC positivity predicted late recurrence in these patients, with a relapse rate of 21% in those women CTC positive versus 2% in those women CTC negative, or, in other words, a relapse rate 13-fold higher in CTC-positive patients. As with bone marrow micro-metastasis, CTCs can be classified by their -biomarker phenotype using immunocytochemistry, PSR, and NGS.

### 6.3. Circulating Tumour DNA

More recently, circulating DNA and RNA also are providing new insights into breast cancer, its prognosis, and therapeutic options. This was recently reviewed by Ho et al. [103]; in summary, circulating tumour DNA (ctDNA) and circulating tumour RNA (ctRNA) form a small fraction of circulating free DNA (cfDNA). cfDNA is released into the blood from cells that are undergoing apoptosis, necrosis, erythyroblastic enucleation, and exosomes. The modifications in the genetic and epigenetic alterations in these molecules can be used to estimate the possible prognosis of a patient, the presence of MRD, options for treatment, and the risk of relapse [104]. The term molecular minimal disease has been coined to define patients who are positive for ctDNA representing a future method for detecting and monitoring patients with breast cancer. Although the levels of ctDNA are low in blood samples, the lower limits of its detection have changed with the development of the polymerase in chain reaction (PCR) and next generation sequencing (NGS) [105,106]. However, the lowest limit of detection (LOD) of ctDNA has not been established or defined [107] and has been recently reviewed [108].

In general, the techniques can be divided into two groups: those that are tumour informed and those that are tumour agnostic. The tumour-informed group are analysed from the biopsy or surgical specimen, firstly using whole exome sequencing (WES) or alternately large gene or tumour-specific panels. From these results, a personalised panel of genes can be determined and is designed for the analysis of blood samples using ultra-depth PCR/NGS [108]. The method detects variants of allele frequency (VAF) with a positive sample having a least two somatic variants detected. By using WES, an estimated positive result was estimated to be 0.042% although the method can detect VAF at levels of 0.002% [109]. With the use of a patient derived panel, the detection of VAFs is reduced to less than 0.01% [109]. Differing from the first group the agnostic group does not analyse the primary tumour but detects ctDNA using a fixed panel of markers or probes; in a study of patients, positive results were defined as the detection of one somatic mutation [109]. More recently, methods using multi-omics such as the methylation of ctDNA or its fragments have improved the sensitivity of the method and a lower limit of detection in patients with colon cancer [110]. In patients with breast cancer, patients after neoadjuvant chemotherapy and who achieved a pathological complete remission had a significant decrease in ctDNA levels [111]. Moreover, it was reported that patients who achieved a pathological complete remission were negative for MRD using this method of detection. Patients who did not achieve a complete pathological response but were ctDNA negative had a similar prognosis as those who did. Meanwhile patients ctDNA positive were significantly at a higher risk of metastatic disease [112]. Those patients who remained ctDNA positive had a significantly higher risk of relapse, metastatic disease, and overall survival [112,113]. The TBCRC 030 trial reported that in TNBC after neoadjuvant chemotherapy the detection of ctDNA was decreased in patients responding to treatment as compared to 24% of those who did not respond to treatment [114].

The detection of ctDNA in plasma samples has not been validated in terms of the VAF and lower cutoff point of detection. However, Janni et al. [115], using a plasma-only multiomic ctDNA technique, could detect ctDNA and if positive was associated with relapse. ctRNA is released in the same way as ctDNA and includes microRNA (miRNA) and long non-coding RNA (IncRNA). The patterns of expressions of miRNAs are tissue specific, being essential for post-transcriptional regulation of gene expression. However, the use of ctRNA as a biomarker is still in an experimental stage [116]. Differing from miRNA, IncRNA plays a role in the regulation of cell growth, differentiation, proliferation, and apoptosis and was recently reviewed [117]. Thus, the role of liquid biopsies offers a new focus on selecting the most optimal therapy [118]. One pitfall of ctDNA is that it does not differentiate between apoptotic cells, necrotic cells, or “living” cancer cells. However, NGS allows the early detection of genetic mutations that may guide treatment options as well determining the progression of MRD to metastatic disease. Theoretically, therapies used to treat metastatic breast cancer could be used in the context of MRD. Using ctDNA to detect mutations that are associated with the use of specific treatments may be carried out. Detection of the mutations of the BRCA, ESR1, HER-2, AKT1, and PTEN have been studied in this context in patients with metastatic breast cancer. Mutations in the ESR1 gene is one of the main causes of endocrine therapy resistance and is associated with a worse prognosis [119]. This mutation causes decreased effectiveness against aromatase inhibitors or the ER modulator tamoxifen [119]. In these cases, a switch to selective oestrogen receptor degraders (SERDs), such as fulvestrant or elacestrant may potentially be more effective [119]. The PADA-1 trial compared standard endocrine therapy versus fulvestrant if there was a rise in ESR1 mutations as detected by ctDNA. Those patients who switched therapy showed a superior progression free survival [120].

Similarly in the PALOMA III phase III trial patients with metastatic breast cancer who had progressed while being treated with endocrine therapy and with ESR-1 and PIK3CA mutations detected using ctDNA compared fulvestrant plus a CDK4/6 inhibitor (palbocicib) versus fulvestrant alone showed an increased progression-free survival. This was seen in patients who when treated with palbociclib with a significant decrease in the expression of this mutation. However, a decrease in the expression of ESR1 was not associated with the progression-free survival [121]. Similarly, in the SOLAR-1 Trail the combination of alpelisib plus fulvestant proved to be superior to fulvestrant alone in patients with or without the PIK3CA mutation [122]; the results of the Bioltal EE trial reported that the combined therapy of letrozole plus ribociclib that in addition to the absence of the ctDNA mutation the clearance of mutation by day 15 was a positive prognostic factor [123]. The ongoing trials have recently been reviewed by Cieslik JP et al. [124].

Newer SERDs have been developed and approved by the FDA and CSM. Elacestrant is a recently oral administered SERD, and ongoing trials are in progress [125]. Palazestrant is another orally administered SERD that has no ER agonist effect and completely blocks the oestrogen-induced transcriptional activity; in addition, it can shrink brain metastasis in mouse models. In preclinical models, it was reported to be superior to fulvestrant in wild-type models, elacestrant in models with the ESR1 mutation, and tamoxifen in patients with cerebral metastasis [126]. In 2024, the FDA approved capovasertib in combination with fulvestrant; a significant benefit in PFS was seen in patients with RE-positive, HER-2-negative who had mutations in the PIK3CA/AKT1/PTEN genes; however, patients who did not harbour these alterations showed an uncertain benefit from combined treatment [127]. Similar results were seen in a Chinese cohort study, the phase 3 CAPItello-291 trial [128]. In those patients with the same mutation pattern, the AKT1 inhibitor capisasertib showed positive effects in patients who had failure of a CDK4/6 inhibitor and anti-ER monotherapy; it also showed less adverse effects than in patients treated with alpelisib [129]. Despite these promising results in metastatic breast cancer, the clinical utility of using ctDNA as a guide to switch therapy regimes is uncertain [130].

Not only can the detection of ctDNA be used to aid treatment options but longitudinal monitoring using this method can detect relapse before clinical relapse is detected [131]. This lead time varies on the subtype of breast cancer but has been reported in all subtypes of breast cancer from 10 months to 2 years in low grade breast cancer [131,132,133,134] and between 4 and 7 months in TNBC [134] and has even been suggested that it potentially replace imaging studies [135]. While micro-metastasis is associated with a worse prognosis, as is the presence of CTCs [94,136] the lead-up time to relapse with respect to micro-metastasis has not been reported, although Sparrano et al. [102] reported a lead time of a median of 2.8 years.

## 7. The Effect of Treatment on the TME and MRD

The 2025 NCCN guidelines on the treatment of breast cancer vary according to the subtype of breast cancer [137]. Depending on the subtype of breast cancer, treatment includes neoadjuvant chemotherapy, surgical excision of the primary tumour, local radiotherapy and adjuvant therapy including chemotherapy, HER2-targeted therapy, and hormonal therapy. Treatment options are based on the analysis of the primary tumour and as mentioned previously MRD may have different biological features, thus treatment selection based on the analysis of the tumour may not be optimal.

## 8. The Latency Period of MRD

The key questions are what causes micro-metastatic dormancy, what triggers the end of dormancy, and finally how to eliminate MRD when present or at least maintaining the latency period? The propensity that micro-metastasis remains in a non-proliferative state for an extended period provides the possible opportunity to maintain these cells in the latent state thus preventing them from becoming active, proliferating, growing, and disseminating to form the metastatic disease.

Mechanisms that trigger dormancy and the end of the latent period in breast cancer patients are largely unknown. The TME and dormancy of tumour cells in different tissues are mediated by signalling from immune cells, stromal cells, and the extracellular matrix. This introduces the possibility of using targeted therapies against MRD, as outlined in the NCCN guidelines, even though trials have been in the treatment of metastatic disease.

What induces dormancy? Intercellular communication between cancer cells and the immune system, lymphocytes, tumour-associated myeloid cells, non-haematopoietic stromal cells impact dormancy and decrease cancer cell proliferation [70]. Exosomes play an important role in promoting cancer cell dormancy. Bone marrow mesenchymal stem cells can suppress BM2 human breast cancer cells in vitro, suppressing their proliferation, decreasing the expression of stem cell-like surface markers, decreasing the cancer cells’ invasive properties, and decreasing their sensitivity to taxanes. The bone marrow mesenchymal stem cells secrete exosomes that contain various miRNAs, especially the over-expression of miR-23b, which induced dormancy in the cancer cells by the MARCKs gene. miR-23b decreases cell cycling and invasive properties by decreasing the expression of MARCKS. The authors suggested that the exosomal transfer of miRNAs may promote dormancy in breast cancer cells [138]. It has also been reported that mesenchymal stem cells release exosomes that are able to transform the cancer cells into dormant cells and impede DNA repair in the tumour cells. These bone marrow mesenchymal stem cells chemotactically migrate towards the implanted cancers cells causing an epigenome reorganisation and this occurs early in the metastatic niche [139].

Less is known about what causes the end of the latency period, it has been hypothesised that changes in the micro-metastasis caused by clonal instability or changes in the TME with an increasing immune dysfunction or a combination of both. MDSCs are one possible candidate and promote cancer cell escape from cancer specific cytotoxic T-cells, escaping from both innate NK-cells and acquired T and B lymphocytes [70]. In prostate cancer bone metastasis models, clusters of tumour cells were found to express myeloid markers, which changed the pathways association with cancer progression and immune modulation. The authors suggested that the fusion of bone marrow cells with the cancer cells may be responsible for these hybrid cells [140]. The most significant changes, as shown by multi-omics were changes in cell adhesion and cancer cell proliferation leading in in vitro models an increased metastatic potential. Using single cell RNA sequencing, it was determined that tumour-associated neutrophils, macrophages, and monocytes were significantly increased in the TME surrounding the hybrid cells, causing an increased immunosuppressive TME. There was enhanced epithelial–mesenchymal transition with more aggressive biological properties and these cells were resistant to docetaxel and ferroptosis while remaining sensitive to radiotherapy [140]. Using a mouse model, Chia et al. reported in those animals infected with the influenza or COVID-19 viruses that pulmonary metastasis grew and that survival rates were lower in infected mice as compared with those acting as controls [141].In this mouse model, it was further shown that the equilibrium of anti-tumour and imunsuppressive immune systems radically changed, with increased formation of epithelial–mesenchymal hybrid cancer cells, which is associated with the end of the latency period. Increased expression of matrix metalloproteinases and IL-6 were also noticeable; IL-6 in inducing an inflammatory microenvironment result in the end of the latency period and cancer cell proliferation and growth and was recently reviewed [141].

Similar results in observational studies showed an increased mortality rate in patients with cancer infected with COVID-19 during the pandemic wave of 2020 and 2021 possibly due to changes in the balance of cytokines [142,143].

It has also been suggested that osteoblasts and osteoclasts affect the dormant cancer cells, in a reversible form, like switching it on or off. Osteoblasts and osteoclasts can regulate this switch both by interacting with the cancer cells directly or by the secretion of cytokines. This switch involved the signalling pathways, TGFβ, Wnt axis, and Notch2 [144]. The TME is dynamic and changes with time; using mouse models, the resorption of bone leads to changes in the equilibrium of the endosteal niche and reactivation of bone micro-metastasis [145]. Periostin and TGFβ1 mediate neo-vasculization of the perivascular niche and frequently terminates the dormancy period of cancer cells [146]. Furthermore, direct communication between tumour and niche cells can lead to a reactivation of dormant tumour cells. It has been suggested that the ECM regulates tumour dormancy by changing the composition of collagen III, its binding to the focal adhesion kinase, which in turn activates the extracellular signal-related kinase (ERK) pathway, which results in the stimulation of mitosis in the cancer cells [147]. In mouse models, inhibition of Wnt induces dormancy and immune invasion and also decreases mitosis in the tumour cells [148].

## 9. Immunosuppression in Breast Cancer

The mechanisms of immunosuppression classify breast cancer as immunologically “cold” with a TME that is essentially immunosuppressive [139]. This immunological TME changes with time, the primary tumour being “colder” than metastatic disease [149]. In metastatic bone disease the immunosuppression mechanisms and lack of cytotoxic cell activation results in a TME immunosuppressive, and immunotherapy is much less effective than in other solid tumours [95]. Thus, the identification of biomarkers that characterise both the molecular, phenotypic, and biological properties that may predict the benefit of immunotherapy in patients with MRD [95].

### 9.1. Targetting the Immunosuppressive Environment of the TME

While the previously mentioned therapies are designed to eliminate or maintain the cancer cells in a latent state, there is increasing research in targeting the immunosuppressive TME. These therapies can be divided into non-specific affecting the whole TME as well as having systemic effects or specific therapies which directly affect CAFs, TAMs, and MDSCs.

### 9.2. The Use of Bisphosphonates

As previously mentioned, MMP-2 plays an important role in the metastatic cascade; however, inhibitors of MMP-2 have been shown to be unacceptably toxic and as such preclinical and clinical trials have been suspended both in first and second generation MMP-2 inhibitors [149,150,151]. Bisphosphonates are used to treat osteopenia, osteoporosis, and hypercalcemia in breast, prostate cancer, and myeloma. The results of the use of bisphosphonates combined with standard adjuvant therapy has shown mixed results. Jallouk el al. [152] reported that there was no benefit of the addition of a bisphosphonate when added to standard adjuvant treatment. However, the Early Breast Cancer Trialists’ Collaborative Group concluded that bisphosphonates significantly reduced relapse and mortality in postmenopusal women whether naturally or medically induced [153]. The benefit of the use of bisphosphonates was independent of receptor status, tumour grade, infiltration of lymph nodes or the use of chemotherapy [154].

Differing from these conclusions, it has been reported that clodronate significantly reduced the risk of both skeletal and visceral metastasis [154,155]. In patients with non-amplified MAF gene expression, clodronate significantly improved disease-free survival and overall survival [155]. Recent updated ASCO-OH (CCO) guidelines, based on a consensus recommendation, say that postmenopausal women (natural or therapy induced) should be treated early with a bisphosphonate, as it showed a modest improvement in overall survival, irrespective of hormone receptor or HER2 status [156].

In addition, bisphosphonates of which zoledronic acid is the most potent, can modulate the immune system via various mechanisms. This change in the immunological balance in the TME may cause beneficial effects with minimal toxicity. Using breast cancer lines in vitro incubation with bisphosphonates produced irreversible inhibition of cancer proliferation, and increased apoptosis and cancer cell necrosis. It was proposed that this was caused by the downregulation of Bcl-2, the release of cytochrome c from mitochondria, and the activation of caspases [157,158,159]. Using the breast cell lines MCF-7, 747D and MDA.MB.231, the effects of clodronate, pamidronate, ibandronate, and zoledronate all produced in a dose dependent manner a significantly decreased proliferation of the MCF-7 and 747D cell lines, although the MDA.MB.231 was more resistant to bisphosphonates. In the MCF-7 and 747D breast cancer cells, apoptosis was induced. In the MCF-7 cell line, this inhibition could be almost completely stopped by caspase inhibition, while in the 747D cell line, inhibition of caspase activity was virtually unchanged [159]. Thus, at least in vitro bisphosphonates could induce cell death that could to a beneficial role in the management of breast cancer.

Zoledronate again was the most potent inhibitor of the invasion of breast cell lines, again in a dose dependent manner but required a lower concentration than that was needed to produce apoptosis [160]. Migration of macrophages is caused by the inhibition of RANK, and bisphosponates reduce the RANK-L-induced dissemination [160]. In breast cancer cells, RANK is also expressed and the inhibition of this pathway may reduce cancer cell dissemination [160]. Zoledronate has a secondary effect in that it causes a repolarization from the M2 to M1 macrophage phenotype using an in vivo rat model [161,162]. Using lipid-coated nanoparticles enveloping calcium zoledronate, it was reported that TAMs were significantly reduced in a S180 mouse model [163,164]. This significantly decreased the immune suppressive TME, decreasing cancer cell growth as well as significantly decreasing angiogenesis without the appearance of systemic adverse effects. In patients with breast cancer, the immunosuppressive TME inhibits NK-cell function by decreasing the expression of NKP46 and NKG2D, which normally activate he NK-cells and increases the inhibitory KIR2DL1 receptors. Using RNA analysis extracted from blood of breast cancer and normal patients and cell cultures it was reported that zoledronate improved the expression of the NK activating receptors [165]. Using MDA-MB-231 cells, zoledronate was able to decrease the activity of Tregs in a dose dependent manner [165]. In a prostate cancer mouse model zoledronate combined with thymosin alpha1 combined with androgen derivation therapy increased the infiltration of CTL into the tumour and increased their cytotoxic activity and stimulated anti-tumour macrophages; whether this occurs in breast cancer is unknown [166]. It also decreased HER2 receptor levels and the downstream signalling pathways in breast cancer; this effect was higher in hormone-resistant cancer cells [167]. Bisphosphonates also decrease the activation of pro-MMP2 by MT-MMP-1 and thus by decreasing MMP-2 expression, the migration and invasion of tumour cells is significantly decreased [168]. Figure 2 shows the effect of FDA approved drugs used in breast cancer.

### 9.3. The Use of All-Transretinoic Acid (ATRA)

Although the mechanisms that cause the differentiation of MDSCs into mature macrophages is not well understood, the use of ATRA significantly decreased the conversion of MDSCs into mature macrophages [169,170]. HF1K16 is a pegylated liposomal form of ATRA, which has a longer half-life and therefore can deliver a higher dosage to the cancer and is currently being investigated in ongoing phase 1 trials [171].

### 9.4. Anti-Cancer Associated Fibroblast Therapies

Targeting the immunosuppressive immunological mechanisms in the TME to increase the immunological effector cells may produce deleterious side effects as seen with the use of PD-1 and PD-L1 check point inhibitors. In patients with a co-existent autoimmune disease, these drugs may exacerbate the disease, and more serious side-effects are interstitial lung disease and myocarditis, which may prove to be fatal [172].

CAFs can be targeted by cytotoxic drugs: talabostat, an oral cytotoxic drug, was reported in mouse models to be able to degrade the extracellular matrix, showing some cancer control [173]. However, in patients with metastatic colorectal cancer it showed no therapeutic effect [173]. The humanised anti-FAP antibody inhibitor, sibrotuzumab, was reported not to have activity against CAFs [174]. CAF cells express FAP, and it has been reported that the use of anti-CAF prodrugs or protoxins when coupled with a FAP cleavage site are systemically administered. However, they are activated by the expression of FAP. Tumour lysis and the inhibition of growth was seen when injected into human breast and prostate xenografts [175,176]. Antibody conjugates targeting FAP using immunotoxins have also been investigated. The anti-FAP-PE39 conjugate suppressed tumour growth and increased the infiltration of tumour infiltrating lymphocytes [177]. Another line of investigation used doxorubicin or anti-Tenacin C liposomes targeted at FAP both to deliver cytotoxic therapy and to remodel the TME [178,179]. The use of a transinfected FAP mRNA dendritic cell vaccine showed a decreased proliferation of cancer cells [180]. A collateral effect was increased NK-cell activity, increased the CTL response, as well as an increased anti-tumoral humoral response [180]. The use of FAP-specific chimeric antigen receptor T-cell therapy has been reported to be able to eliminate most FAP positive CAFs and decrease tumour stromal generation; however, significant side effects were observed, especially with respect to bone marrow toxicity [181]. However, none of these strategies has been studied in clinical trials due to efficacy and safety side effects. Pirfenidone is an oral anti-fibrotic drug, used to treat idiopathic lung fibrosis. It has also been reported to target CAFs, producing a restriction of tumour cell proliferation, the immunosuppressive TME, the formation of metastasis, resistance to anti-tumour treatment, and extracellular matrix stiffness. Used in combination with doxorubicin, it produces synergistic anti-tumour effects while not damaging normal tissues [182].

The reduction in the number of CAFs has positive collateral effects, such as the reduction in MDSC recruitment into the TME, a decrease in TAMs, decreased intra-tumoral recruitment and survival of Tregs, as well as decreasing the secretion of cytokines and chemokines that favour the immunosuppressive environment [182].

### 9.5. Tumour Associated Macrophages

Circulating monocytes are the precursors of TAMs and due to their plasticity can infiltrate the TME where they form up to 50% of the TME [183]. Their phenotype is determined by the TME converting them into the M2 immunosuppressive subtype rather than the anti-tumour M1 subtype [184]. Thus, they play a function in the immunosuppression of the patient’s immunological system, promote cancer cell proliferation, angiogenesis, and finally metastasis [185]. The is a metabolic dialogue between the tumour cells and TAMs via exosomes, which causes metabolic changes in the TME, enhancing tumour cell survival [186,187]. The Warburg effect is the upregulation of aerobic metabolism by tumour cells, which inhibits apoptosis and increases their survival rate, increasing the production of lactic acid, and this increases the TAMs into the M2 phenotype [184]. Tumour cells also affect lipid metabolism with an increased need to produce lipids, biofilm construction and function maintenance. It also increases the conversion of TAMs into the M2 phenotype [188]. In addition, the increased levels of lactic acid promote the Warburg effect increasing TAMs into the M2 phenotype [184]. Thus, drugs that affect both tumour cell and TAM metabolism may aid in decreasing the immunosuppressive TME. Treatment with an inhibitor of lactate dehydrogenase can significantly improve the effect of paclitaxel and trastuzumab against resistant cancer cells, with Galloflavin and Machilin A being two examples [189,190]. It has been reported that simvastatin, a cholesterol-lowering drug, acts on the TAMs rather than the tumour cells, reverses the epithelial–mesenchymal-transition and promotes the conversion of TAMs to the M1 phenotype [191]. It has also been reported that chloroquine resets the Tam population to the M1 phenotype thus decreasing the immunosuppression of the TME [192]. PARP inhibitors also reprogram the metabolic features of the TME, activating M1 TAMs and CTL and was extensively reviewed by Liang et al. [193]. Tumour progression is accompanied by fibrosis due to the activity of CAFs resulting in the stiffening of the extracellular matrix. This change in the extracellular matrix decreases the ability of immune effector cells from entering the tumour. It has been reported that TAMs, because of this stiffening, initiate the biosynthesis of collagen directed by transforming growth factor ß [194]. This changes the metabolic environment causing CTL exhaustion and an adverse TME for CTLs [194]. In the TME, TAMs that do not express PD-L1 are associated with immunosuppression, and this allows them to co-localise with cancers. Inversely, TAMs that do express PD-L1 stimulate the anti-tumour immune system. During the monocyte to macrophage maturation PD-L1 is upregulated; these TAMs are mature cells and can stimulate CTL proliferation and their cytotoxic efficacy and thus are a good prognostic marker of outcome in breast cancer patients [195].

### 9.6. Myeloid Derived Suppressor Cells

MDSCs play a pivotal role in creating the immunosuppressive environment of the TME, as previously mentioned. Contrasting with physiological maturation of haemopoietic cells, the TME renders MDSCs incapable of differentiating into mature cells which results in a heterogeneous cell population [196]. TMEs with high concentrations of MDSCs are associated with a worse prognosis and decreased responses to immunotherapy [197]. The proliferation of MDSCs is driven mainly by tumour derived growth factors, including GM-CSF, G-CSF, M-CSF, VEGR, and IL-6 [198]. In a mouse model, the selective JAK/STAT3 inhibitor JSI-124 was able to reduce the number of MDSCs by promoting their differentiation [199]. However, in tumour sites the activity of STAT3 is lower than in the blood or spleen and JSI-124 caused no significant change in number of MDSCs in the TME [199]. The immunosuppressive effect of MDSCs in the TME requires the activation by cytokines produced by tumour stromal cells or activated T-cells. These activated MDSCs in mouse models and they specifically and effectively decrease CD8^+^ function by decreasing the levels of IFN gamma [200]. By activating the apoptotic cascade of T-cells, the MDSCs reduce the generation of T-cells and thereby promote immunosuppression through immune blocking rather than directly killing the T-cells [201] through a high expression of ARG1, iNOS, and ROS in MDSC [202,203]. MDSCs show a great deal of plasticity in their biological properties, being able to stimulate the production of Tregs and inhibiting CTL activation [204,205], thus significantly reducing the effectivity of immunotherapy [205]. Targeting MDSCs is therefore a focus of research on how to eliminate MDSCs and decrease the immunosuppressive TME.

In the context of MRD, current lines of investigation to eliminate MDSCs are firstly the CD33 targeted antibody-conjugate gemtuzumab ozogamicin is approved for use in patients with CD33^+^ acute myeloid leukaemia. It targets CD33 on the cell membrane and in doing so releases a derivative of the cytotoxic agent calicheamicin which affect internalisation into the tumour cell, causing cell death [206]. Independent of the subtype of MDSCs membrane, CD33 is found to be expressed in all MDSCs and as a result increases the death of these cancer cells; thus, it raises the question of it being effective in breast cancer MDSCs [207]. A second alternative is the differentiation of MDSCs into mature macrophages, although the mechanisms that regulate this is not well understood at present, the use of all transretinoic acid (ATRA) has already been mentioned, significantly decreasing the presence of MDSCs into mature macrophages. Thirdly, there is the possibility of blocking the immunosuppression effect of MDSCs. In models on mouse breast cancer, phosphodiesterase-5 inhibitors such as sildenafil, tadalafil, and vardenafil reversed the immunosuppressive effects of MDSCs and thus improve anti-tumour immune response [208]. The use of histone deacetylase inhibitors is another type of therapy. The use of entinostat decreased the frequency of circulating MDSCs [209] and increased the efficacity of immune checkpoint inhibitors [210]. A new approach, at least in mice models, is the use of self-assembled gemcitabine-celecoxib nano-twin drug therapy. The nanoparticles containing both drugs have a log half-life in the circulation enabling them to accumulate preferentially in the TME. The release of both drugs acts synergistically, in both inhibiting the proliferation and increasing apoptosis in Tregs, depleting MDSCs, activating both CTL and NK-cells, and converting TAMs to the M1 phenotype. This combination in mice with 4T1 breast tumours showed an increase in the anti-cancer and anti-metastasis capabilities as compared to placebo [211]. Figure 3 summarises these pre-clinical data.

## 10. Where Are We Now and Where Should We Be Going?

For more than a century, treatment of cancer, including breast cancer, has been based on the eradication of cancer cells. With the new development of different technologies such as ER and HER-2 status and the phenotypic and biological properties of the breast cancer cells, new therapeutic options have been developed, many in the use of the metastatic setting. It has also allowed the subclassification of breast cancer and, depending on these characteristics’ treatments may be different, the idea of precision medicine [3]. However, this is a one-sided ball game, whereby breast cancer cells only form a small part of the TME. The cancer cells form part of the stromal tissue whereby direct cell to cell communication, exosomes, cytokines, chemokines, and enzymes such as the matrix metalloproteinases reshape the TME to promote an immunosuppressive environment. This has important consequences, leading to a decreased anti-tumour immune response. The TME is in a dynamic state and evolves with time; this ecosystem or onco-sphere may differ between the primary tumour and distant micro-metastatic disease, depending on the tissue invaded. These different phenotypic and biological characteristics of the cancer cells may require different therapies, which at present is the not the case, as metastatic disease detected in imaging studies is rarely biopsied. These changes in the TME may be partly due to the clonal instability of the cancer cells, the effect of treatment, and/or changes in the immunological equilibrium of immunosuppression and the anti-tumour immune response. During the last decade, there has been increasing interest in the TME and its interactions with the tumour cells to change treatments into a two-balled game. Thus, the published findings may in the future change treatment options to decrease the immunosuppressive effects of the TME and to improve the anti-tumour immune response.

Initial studies used in vitro techniques using breast cancer cell lines of different metastatic potential; later in vivo studies used genetically modified mouse models. Not only did this provide new insights into the TME but also new treatment options even though they are in their infancy with few phase III trials being reported.

## 11. Conclusions

The presence of MRD after curative treatment will ultimately determine if the treated patient will relapse in the future. Although it is important to develop new therapies to eliminate residual breast cancers, it is possible that targeting the immunosuppressive environment to increase the anti-tumour response may improve the outcomes in patients with breast cancer. Therefore, additional studies are required, not only in the pretext of animal models but also understanding the complex TME, and thus at least prolonging the latency period. This involves targeting the oncosphere as one entity either with combined or sequential therapies.

## Figures and Tables

**Figure 1 ijms-26-11346-f001:**
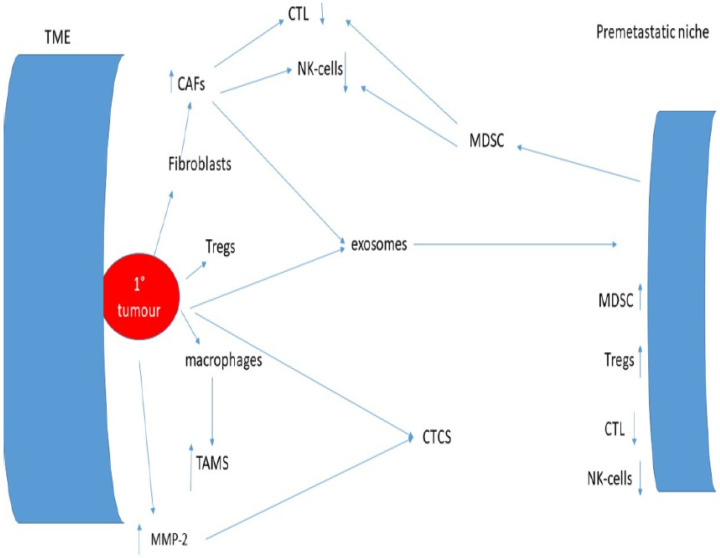
Interactions between the tumour microenvironment, the immune system, and the pre-metastatic niche.

**Figure 2 ijms-26-11346-f002:**
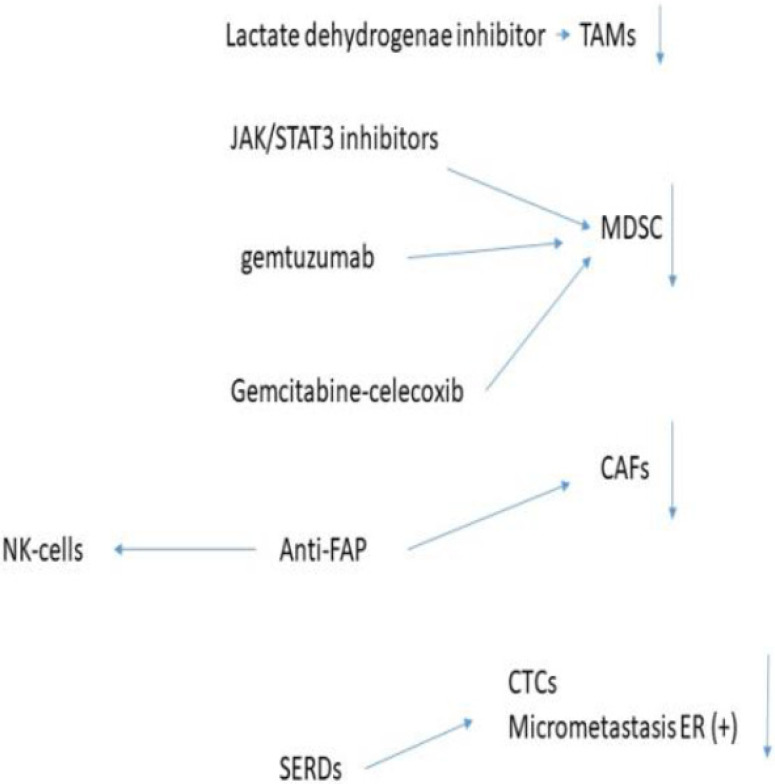
Effect of FDA approved drugs in breast cancer.

**Figure 3 ijms-26-11346-f003:**
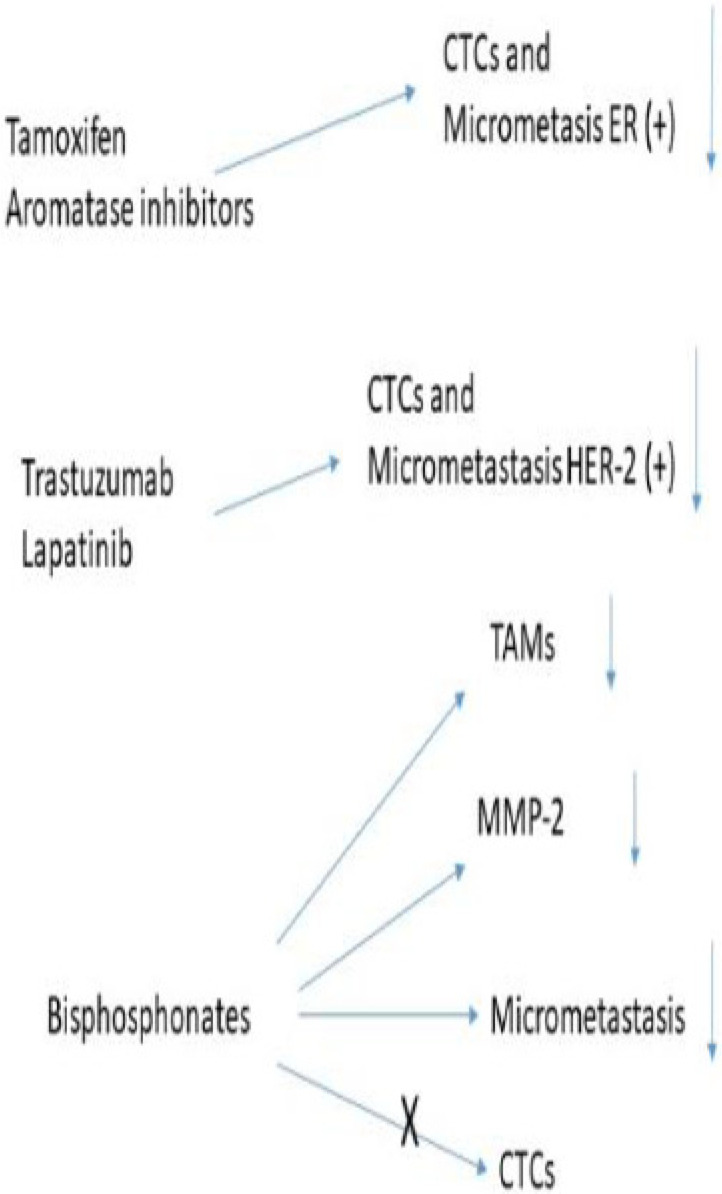
Drugs approved by the FDA to treat breast cancer.

## Data Availability

No new data were created or analysed in this study. Data sharing is not applicable to this article.

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
