# Peer review of "Minimal Residual Disease in Breast Cancer: Tumour Microenvironment Interactions, Detection Methods and Therapeutic Approaches"

_ijms, 2025, doi:10.3390/ijms262311346_

Round 1
Reviewer 1 Report
Comments and Suggestions for Authors
In this review manuscript “ Minimal Residual Disease in Breast Cancer, methods of its detection, treatment and can it prevent metastasis?”, the authors comprehensively explore the role of minimal residual disease (MRD) in breast cancer, addressing its biological basis within the immunosuppressive tumour microenvironment (TME), methods of detection (including bone marrow micro-metastasis, circulating tumour cells (CTCs), and circulating tumour DNA (ctDNA), and diverse therapeutic strategies targeting MRD to prevent metastasis. The work critically synthesizes recent advances in immunotherapy, antibody-drug conjugates (ADCs), and microenvironment-modulating agents. Here are specific comments and suggestions:
Major comments:
- The abstract lacks mention of metastasis prevention, your main point. So you can add some details to make reader get your opinions easier.
- Conclusion summarizes mechanisms but fails to explicitly answer whether MRD-targeted treatment can prevent metastasis, despite extensive therapeutic reviews. It’s the main question ofthe manuscript, so a direct evidence is needed.
- The “Subclassification of MRD” section introduces vital prognostic MRD subtypes (Fig 5). This framework should be integrated when evaluating therapies, such as “Immunotherapy Using Monoclonal Antibodies”, “Targeting CAFs/MDSCs/TAMs”.
Minor comments:
- The “Conclusions” should briefly acknowledge limitations, such as preclinical data reliance for novel therapies and variability in detection methods.
- You can refine your structure of the article to make it more clear. Consider consolidating related topics.
Author Response
Firstly, we would like to thank you for your time and valuable recommendations to improve the article. All the corrections are in red. As some of your points are the same, we have combined our response in these cases.
Reviewer 1: We completely agree with your comments on the abstract and this has been fully rewritten.
Reviewer 1: As for the conclusions this has also been rewritten, to try to answer the question of whether MRD targeted therapy can prevent metastasis. There is very little literature, mainly from preclinical animal model studies. However, we have included the subject in the conclusions and the articles limitations.
Reviewer 1: All figures and diagrams have been deleted; we think that they do not contribute to the articles relevance and cause more complications than benefits.
Reviewer 1: In the section “subclassification of MRD” the same framework has been integrated when evaluating therapies directed at CAFs, MDSCs, TAMs and monoclonal antibodies
Reviewer 1: The article has been re-structured, as you rightly point out to make it easier to follow and make it more clear
Reviewer 2 Report
Comments and Suggestions for Authors
- It is unclear what the focus of this review is. Based on the title, one may assume that the review is focused on MRD of breast cancer. However, the abstract does not mentioned MRD at all but rather present a review focused on the tumor microenvironment. The manuscript itself does contain sections that cover both the TME and MRD, but without significant connection linking the two fields. I would recommend focusing the review on one of the subjects.
- Figure 3 is presented in a very confusing fashion and it is unclear what the message is from this figure.
- The organization and subtitles of the manuscript is quite confusing. For example, Section 3-7 should fall under section 2 as sub-sections.
- Figure 5 is a direct copy of a published figure from another paper. Even though citation is given in the figure legend, this is highly inappropriate.
- Section 10 is way too short and does not actually contain any information relevant to the title of the section at all. It is possible that the authors meant to include a few following sections as sub-sections but it is organizationally unclear.
- Section 14-21, authors discussed many of the current therapeutic modalities for treating breast cancers. However, there was not sufficient information and focus given on how these therapies affect either MRD or TME. I would recommend the authors to omit the general information regarding the mechanism of action and therapeutic benefit of these therapies but rather focus on their impact on the TME/MRD, depending on which one will be the focus of this review.
- It is generally argued in this review that the dysfunction of the immune system and the “cold” immunological nature of breast cancer that allows for MRD’s evasion and circulation. It is worth noting that recent literature suggests inflammation/immune activation as a critical trigger of the re-awakening of dormant breast cancer MRD (Chia et al., Nature, 2025).
- Section 22-28 appear to be the actual focus of the review. However, this section covered the effects of therapy on both TME and MRD without much information connecting the two subjects and the authors bounced around between the two topics. I would again suggest focusing the review on one of the subjects and establish a clearer structure.
Author Response
Dear Reviewers
Firstly, we would like to thank you for your time and valuable recommendations to improve the article. All the corrections are in red. As some of your points are the same, we have combined our response in these cases.
Reviewer 2: We are completely in agreement and the abstract has been completely re-written. Cancer cells interact with the TME in the form of direct cell to cell crosstalk, via exosomes and cytokines and chemokines, to create an immunosuppressive TME and reducing the anti-tumour immune response. They are interlinked and cannot be separated, we have addressed this recommendation by linking the two aspects, the effects of cancer cells and the changes in the TME they produce along with how these changes in the TME affect the hosts immune response.
Reviewer 2: All figures and diagrams have been deleted as we feel that they do not contribute to the article and cause more complications that benefits.
Reviewer 2: The article has been re-written to address your recommendations 3 and 5. Similarly with regards to your sixth recommendation, all the current therapeutic options for the treatment of primary and metastatic breast cancer have been deleted. The persistence of MRD after “curative therapy” implies that the cancer cells as part of the TME were resistant and so this information does not contribute to the articles main aim.
Reviewer 2: We have included your recommendation that inflammatory changes such as after viral infections may activate the TME and cause an end to the latency period.
Reviewer 2. Point 8 has been addressed, cancer cells interact with the TME in the form of direct cell to cell crosstalk, via exosomes and cytokines and chemokines, to create an immunosuppressive TME and reducing the anti-tumour immune response. They are interlinked and cannot be separated, we have addressed this recommendation by linking the two aspects, the effects of cancer cells and the changes in the TME they produce along with how these changes in the TME affect the hosts immune response. Limitations of this review is that much of the evidence is from preclinical models, phase I and II trials have been stopped either for toxic side effects or no significant differences between treatment and placebo, except in the case of bisphosphonate therapy, there needs to be randomized double blind phase III, multicentre studies to finally answer this question
Round 2
Reviewer 2 Report
Comments and Suggestions for Authors
I would like to to thank the authors for their effort in improving this manuscript. After the revision, the current manuscript presents a much more coherent and clear structure regarding the topic between MRD interaction with TME.
One minor note: currently, section 13.2 appears to be blank, which I believe to be an editing error by the authors. Please correct this issue before publication.
Author Response
Comment1:
I would like to to thank the authors for their effort in improving this manuscript. After the revision, the current manuscript presents a much more coherent and clear structure regarding the topic between MRD interaction with TME.
One minor note: currently, section 13.2 appears to be blank, which I believe to be an editing error by the authors. Please correct this issue before publication.
Reply: The numbering of section 13 was my fault and has been corrected.